# Genetic associations between genes in the renin-angiotensin-aldosterone system and renal disease: a systematic review and meta-analysis

Laura Jane Smyth,[1] Marisa Cañadas-Garre,[1] Ruaidhri C Cappa,[1] Alexander P Maxwell,[1,2] Amy Jayne McKnight[1]

¹Epidemiology and Public Health Research Group, Queen's University Belfast Centre for Public Health, Belfast, UK
²Regional Nephrology Unit, Belfast City Hospital, Belfast, UK

**Correspondence to**
Dr Laura Jane Smyth;
laura.smyth@qub.ac.uk

## ABSTRACT

**Background** Chronic kidney disease (CKD) is defined by abnormalities in kidney structure and/or function present for more than 3 months. Worldwide, both the incidence and prevalence rates of CKD are increasing. The renin-angiotensin-aldosterone system (RAAS) regulates fluid and electrolyte balance through the kidney. RAAS activation is associated with hypertension, which is directly implicated in causation and progression of CKD. RAAS blockade, using drugs targeting individual RAAS mediators and receptors, has proven to be renoprotective.

**Objectives** To assess genomic variants present within RAAS genes, *ACE*, *ACE2*, *AGT*, *AGTR1*, *AGTR2* and *REN*, for association with CKD.

**Design and data sources** A systematic review and meta-analysis of observational research was performed to evaluate the RAAS gene polymorphisms in CKD using both PubMed and Web of Science databases with publication date between the inception of each database and 31 December 2018. Eligible articles included case–control studies of a defined kidney disease and included genotype counts.

**Eligibility criteria** Any paper was removed from the analysis if it was not written in English or Spanish, was a non-human study, was a paediatric study, was not a case–control study, did not have a renal disease phenotype, did not include data for the genes, was a gene expression-based study or had a pharmaceutical drug focus.

**Results** A total of 3531 studies were identified, 114 of which met the inclusion criteria. Genetic variants reported in at least three independent publications for populations with the same ethnicity were determined and quantitative analyses performed. Three variants returned significant results in populations with different ethnicities at p<0.05: *ACE* insertion, *AGT* rs699-T allele and *AGTR1* rs5186-A allele; each variant was associated with a reduced risk of CKD development.

**Conclusions** Further biological pathway and functional analyses of the RAAS gene polymorphisms will help define how variation in components of the RAAS pathway contributes to CKD.

## Strengths and limitations of this study

► Individuals with microalbuminuria were excluded from both the case and control definitions to improve the discrimination between individuals with and without renal disease as microalbuminuria may regress, remain stable or progress to macroalbuminuria.

► Due to previously reported heterogeneity between different ethnic groups, we included this as a risk factor and performed each analysis per ethnicity.

► Some studies in our search could not be included in quantitative analysis as they lacked information relating to genotype counts and had an unclear measure and definition of albuminuria for both cases and controls.

as a reduction in glomerular filtration rate (GFR) to <60 mL/min/1.73 m², or the presence of persistent urinary abnormalities including albuminuria and/or structural alterations which have been present for at least 3 months.[1] CKD is an increasing public health issue given its associated morbidities, premature mortality and management of advanced CKD is a significant burden on healthcare budgets worldwide.[2 3] There is substantial evidence that inherited genetic variants,[4] the presence of diabetes[5] or hypertension[6] and an individual's ethnicity[7 8] directly influence the development of various CKD phenotypes.

The renin-angiotensin-aldosterone system (RAAS) is a homeostatic endocrine system which is of critical importance to the regulation of blood pressure and maintenance of fluid and electrolyte balance.[9 10] Renin (REN), secreted from the juxtaglomerular apparatus in response to reduced renal perfusion pressure, catalyses the conversion of angiotensinogen (AGT) to angiotensin I.[11] Subsequently, angiotensin I converting

## INTRODUCTION

Chronic kidney disease (CKD) is defined as a progressive loss of renal function measured

enzyme (*ACE*) cleaves angiotensin I to generate angiotensin II, which regulates heart and kidney function by binding to and activating angiotensin II receptors (type I and type II).[11 12] The angiotensin II type I receptors are responsible for multiple biological actions of RAAS including vasoconstriction and sodium reabsorption.[11–14]

Increased RAAS activation is linked to progression of CKD of different aetiologies, especially diabetic nephropathy (DN),[11 15–18] and is mediated by hypertensive injury[17] and accelerated renal fibrosis.[19] The physiological relevance of this pathway in the progression of CKD has focused attention on RAAS components including *ACE*, *ACE2*, *AGT*, angiotensin II receptor type 1 (*AGTR1*), angiotensin II receptor type 2 (*AGTR2*) and renin (*REN*), as candidate genes for various CKD-related phenotypes. Multiple studies have implicated RAAS gene variants in the progression of CKD.[20–26]

This manuscript describes a systematic review and meta-analysis to examine published data reporting genetic variants present within six of the RAAS candidate genes: *ACE*, *ACE2*, *AGT*, *AGTR1*, *AGTR2* and *REN*, for a range of CKD phenotypes and ethnicities, to help define their putative roles as risk factors for CKD.

## METHODS
### Search strategy
A systematic search was undertaken following recognised methods, the Meta-analysis of Observational Studies in Epidemiology (MOOSE) guidelines,[27] by two investigators. PubMed and Web of Science online databases were searched for studies published between the inception of each database and 31 December 2018. All search terms are detailed in online supplementary table S1.

The general strategy of the searches was to follow the structure: (all alternative versions of each gene name, separated by the Boolean operator OR) AND (kidney OR nephrology OR nephropathy OR renal) AND (SNP OR polymorphism OR variant OR allele OR genotype). For each search term, where an appropriate Medical Subject Headings (MeSH) term was available, the query included the quoted search term OR the MeSH term. Additional filters including English OR Spanish languages, human studies, case–control studies, not clinical trials, not review articles, not a case report and not a meta-analysis were applied.

Reference lists from included publications were also manually searched. Two authors (LJS and MCG) independently conducted the literature search, screened the articles and extracted the data. In the case of any disagreement, a third author (RCC) considered the articles. A range of CKD phenotypes were included in this analysis, the case and control definitions are included in table 1.

### Inclusion/exclusion criteria
Inclusion criteria were judged against a standardised list of agreed criteria (LJS, MCG and AJM), where the English or Spanish language publication described an

adult, human case–control study of a defined kidney disease. In the rare instances when suspected duplicate data were identified within two or more included articles, only the article either published first, or that with the larger number of participants was included.

Articles were excluded if they included paediatric subjects, were a pharmacological-based study reporting clinical trials of medications, did not contain genotypic data for the correct gene or were non-human studies.

**Table 1** Phenotypic comparisons included in this analysis

| Case group | Control group |
| --- | --- |
| Autosomal dominant polycystic kidney disease | Healthy controls |
| Atherosclerotic renal artery stenosis | Healthy controls |
| Balkan endemic nephropathy | Healthy controls |
| Chronic glomerulonephritis | Healthy controls |
| Chronic kidney disease | Healthy controls |
| Diabetic nephropathy* | Diabetes mellitus |
| Diabetic nephropathy* | Healthy controls |
| End-stage renal disease | Healthy controls |
| End-stage renal disease | Type 1 diabetes mellitus |
| Focal segmental glomerulosclerosis | Healthy controls |
| Glomerulonephritis | Healthy controls |
| Hypertension-related renal disease | Healthy controls with hypertension |
| Hypertension-related renal disease | Healthy controls |
| IgA nephropathy | Healthy controls |
| Interstitial nephritis | Healthy controls |
| Lupus nephritis | Systemic lupus erythematosus |
| Lupus nephritis | Healthy controls |
| Minimal change nephrotic syndrome | Healthy controls |
| Non-Balkan endemic nephropathy | Healthy controls |
| Nephroangiosclerosis | Healthy controls |
| Polycystic kidney disease | Healthy controls |
| Primary membranous glomerulonephritis | Healthy controls |
| Primary membranous glomerulonephritis | Organ donors |
| Renal transplant recipients | Healthy controls |
| Renal transplant recipients | Kidney donors |
| Type 1 diabetic nephropathy* | Type 1 diabetes mellitus |
| Type 1 diabetic nephropathy linked to end-stage renal disease | Healthy controls |
| Type 1 diabetic nephropathy linked to end-stage renal disease | Type 1 diabetes mellitus |
| Type 2 diabetic nephropathy* | Type 2 diabetes mellitus |
| Type 2 diabetic nephropathy linked to end-stage renal disease | Healthy controls |
| Type 2 diabetic nephropathy linked to end-stage renal disease | Type 2 diabetes mellitus |

*In the studies including diabetic nephropathy as cases, only individuals with reported macroalbuminuria or proteinuria were included. Individuals with microalbuminuria were excluded.

## Data extraction

Where available, the size of each study, case group disease definition and number of individuals, control group definition and number of individuals, ethnicity, genetic variant, genotype in the format of allele 1–heterozygote–allele 2 and allele counts were recorded and calculated in spreadsheets by two authors (LJS and MCG). Articles were reassessed where any disagreement occurred and a third reviewer was employed. Ethnicities were recorded from the articles and recoded following the International Genome Sample Resources' online guidance.[28] Where any population did not align to a listed population code, a new one was created for the purposes of this study. All ethnicity codes are available in online supplementary table S2. The data collected were divided into disease phenotype groups to ensure a high level of homogeneity.

## Statistical analysis

Each genetic variant which had been investigated and reported in at least three independent publications for the same ethnicity and phenotype was included in quantitative statistical analyses. Review Manager (RevMan V.5.3) (The Cochrane Collaboration, The Nordic Cochrane Centre, Copenhagen, Denmark) was employed to facilitate the analysis of allele frequencies. For each single nucleotide polymorphism (SNP), the total number of alleles was recorded per case and control group. Hardy-Weinberg equilibrium was calculated for all included studies, for cases and controls separately. Statistical analyses were performed using the random-effects model as heterogeneity was expected. For each SNP, this analysis provided the p value, OR and 95% CIs. It also facilitated the assessment of the heterogeneity level using the $I^2$ statistic.[29] Forest plots and funnel plots were generated automatically to assess publication bias and the significance value was set at p<0.05 (LJS and RCC).

The genotyping quality of the studies was assessed by reported genotype completion rate and Hardy-Weinberg equilibrium. Phenotypes included in this analysis are shown in table 1; individuals with microalbuminuria were excluded alongside the studies that focused on disease progression. No sensitivity analysis was performed. All study methodologies conformed to the MOOSE criteria.[27] No published protocol is available for this review. The workflow followed a consistent pattern for each gene. A summary of this is included in figure 1.

## Patient and public involvement

Neither patients nor the public were directly involved in the design of this study, which analysed previously published data available in the public domain.

## RESULTS

The database searches returned 3531 results, 144 of which remained following the application of inclusion and exclusion criteria and the removal of any SNP which was not reported on at least three occasions.

Several articles included data for more than one gene, signifying that they have emerged multiple times throughout the database searches; the total number of individual articles was therefore 114. The search strategies are included as online supplementary figure S1A–F. All excluded studies are listed in online supplementary table S3A–F.

The total number of subjects analysed within these studies (n=114) was 18 231 individuals with renal disease and 21 887 individuals acting as controls. For SNPs in three of the RAAS genes, *ACE2*, *AGTR2* and *REN*, there were less than three independent populations studied. A summary table detailing the main results is included as table 2.

## Angiotensin converting enzyme

A total of 15 quantitative analyses were completed for the insertion/deletion (I/D) polymorphism located within *ACE* in eight phenotypes, details of which are included in online supplementary table S4. Three quantitative analyses returned a significant result. The first analysis comprised 11 publications, each studying an East Asian population with type 2 diabetes and nephropathy (T2DN) and compared with type 2 diabetes mellitus without nephropathy (T2DM). Figure 2A displays these results, p=0.009; OR 0.74; 95% CI 0.59 to 0.93, $I^2$=55%, showing that the presence of the insertion variant at this *ACE* locus was significantly associated with this phenotype. The insertion provides a lower risk of developing T2DN in an East Asian population as demonstrated in five studies, which contributed to 51.5% of the weight in this analysis. Figure 2B shows the associated funnel plot for this analysis.

The *ACE* insertion variant was similarly significantly associated with a lower risk of T2DN compared with T2DM in a South Asian population, despite the presence of a high level of heterogeneity (p=0.01; OR 0.57; 95% CI 0.37 to 0.87; $I^2$=89%). This was consistent with the direction of effect in four studies, which contributed to 67.7% of the weight in this analysis (figure 2C). The funnel plot for this analysis is displayed within figure 2D.

The comparison of East Asian individuals with end-stage renal disease (ESRD) compared to a healthy population with no evidence of renal disease also showed a significant association with moderate levels of heterogeneity (p=0.08; OR 0.8; 95% CI 0.67 to 0.94; $I^2$ 68%). Four studies, supporting 69.7% of the weight in the analysis had shown this effect (figure 2e). Figure 2f shows the associated funnel plot for this analysis.

In each analysis (T2DN vs T2DM; end-stage renal disease [ESRD] vs normal), the presence of the *ACE* insertion was associated with a lower risk of developing the CKD phenotype in the respective populations. The non-significant forest plots are included in online supplementary figure S2A–I and the associated funnel plots within online supplementary figure S3A–I.

**Electronic Database Search**

PubMed and Web of Science Databases were searched yielding 3,531 results in total for RAAS genes (*ACE, ACE2, AGT, AGTR1, AGTR2* and *REN*)

↓

**Duplicate Papers Removed**

419 duplicate papers were removed from each gene search. Duplicates remained between genes as several articles contained data for several genes.

↓

**Abstract Screening**

2,888 articles were removed as they did not meet the inclusion criteria.*

↓

**Full Text Screening**

50 articles were removed as they included overlaps in patient groups published elsewhere or contained no data.

↓

**Inclusion of Articles Identified from Bibliographies**

53 papers were identified from bibliographies during full-text screening.

↓

**Final Study Selection for Data Extraction**

529** papers met the inclusion criteria for all genes and were included for data extraction. 227 individual articles.

↓

**Final Study Selection for Quantitative Analysis**

83 articles were removed as the data provided did not generate at least three populations with the same ethnicity per SNP. 144 papers were included in the quantitative analysis.

*Inclusion criteria: Any paper was removed from the analysis if it was not written in English or Spanish, was a non-human study, was a paediatric study, was not a case-control study, did not have a renal disease phenotype, did not include data for the gene of interest, was a gene expression based study or had a pharmaceutical drug focus.

**Final selection: Several papers provided data for > 1 gene, so this figure includes duplicates.

**Figure 1** Workflow pattern. RAAS, renin-angiotensin-aldosterone system; SNP, single nucleotide polymorphism.

### Angiotensinogen

Seven quantitative analyses were completed for rs699, where the T allele was compared with the C allele. Details of each of these comparisons are included in online supplementary table S5. One significant result was identified—the comparison of ESRD with healthy controls in a European population. The results were (p=0.002; OR 0.84; 95% CI 0.76 to 0.94; $I^2$=0%) and

are shown in figure 2G. Only the study with population size over 2000 patients, supporting 46.2% of the weight in the meta-analysis, had achieved significance with the original figures. These results indicate that the presence of the T allele is associated with a lower risk of developing ESRD in this population. The funnel plot for this analysis is displayed within figure 2H. The forest plots containing the non-significant results are included in

**Table 2** Summary of the most significant result for each included gene

| Gene | Articles returned (n) | Articles analysed (n) | Maximum individuals in analysis (n) | Most significant result | P value | OR (95% CI) | I² (%) | Allele | Average allele frequency–controls |
|---|---|---|---|---|---|---|---|---|---|
| ACE | 380 | 95 | 33247 | I/D (EAS) | 0.008 | 0.80 (0.67 to 0.94) | 68 | Insertion | 0.62 |
| ACE2 | 1556 | 0 | NA | NA | NA | NA | NA | NA | NA |
| AGT | 693 | 33 | 13234 | rs699 (EUR) | 0.002 | 0.84 (0.76 to 0.94) | 0 | T | 0.57 |
| AGTR1 | 200 | 16 | 6917 | rs5186 (SAS) | 0.001 | 0.71 (0.58 to 0.87) | 37 | A | 0.81 |
| AGTR2 | 29 | 0 | NA | NA | NA | NA | NA | NA | NA |
| REN | 673 | 0 | NA | NA | NA | NA | NA | NA | NA |

EAS, East Asian; EUR, European; I/D, insertion/deletion; NA, not applicable; SAS, South Asian.

online supplementary figure S4A–F, and the associated funnel plots within online supplementary figure S5A–F.

### Angiotensin II receptor type 1

Four quantitative analyses were performed for *AGTR1* rs5186, comparing the A allele with the C. Details of these comparisons are included within online supplementary table S6. One of the four studies returned a significant result for association, the comparison of T2DN with T2DM in a South Asian population (p=0.001; OR 0.71; 95% CI 0.58 to 0.87; $I^2$=37%). This result resembled the findings of two of the three studies which contributed 87% of the weight in the meta-analysis, as shown in figure 2I. Figure 2J contains the associated funnel plot. These results indicate that the presence of the A allele is associated with a lower risk of developing T2DN in this population. The non-significant results in forest plot format are included in online supplementary figure S6A–C and in funnel plot format within online supplementary figure S7A–C.

Studies not complying with Hardy-Weinberg equilibrium are shown in online supplementary table S7. Allele frequencies for healthy control populations included in this analysis were assessed. None showed signs of bias in any included study, in comparison to all available dbSNP reported frequencies as shown in online supplementary file S1.

### DISCUSSION

Investigations into RAAS genetic variants previously reported to have been associated with a range of CKD phenotypes were undertaken. A total of 3531 studies were identified, 114 of which met the inclusion criteria. Subsequently, 26 quantitative analyses were completed for three RAAS genes where there were at least three independent population studies of the RAAS gene variants. Five significant results within three genes were obtained at the significance level p<0.05, each revealing an association with CKD.

*ACE* is encoded by *DCP1* and is a key component of the RAAS. It catalyses the modification of angiotensin I to II, which is more biologically active.[30] *ACE* is the most frequently studied gene of the RAAS.[31] First sequenced in 1992,[32] this 287 bp Alu repetitive element at intron 16[32] is located on chromosome 17 and is represented by four individual SNPs: rs4646994, rs1799752, rs4340 and rs13447447. Further understanding of its genetic architecture and disease associations may enable patient groups to benefit from targeted therapies with ACE inhibitors.[33]

Since 1994, the association of *ACE* and DN has been rigorously investigated,[33 34] with studies returning conflicting results. A meta-analysis undertaken in 2005 by Ng and colleagues[35] reported a statistically significant result wherein the *ACE* insertion was associated with protection from development of DN in Asians and Caucasians. A second meta-analysis undertaken in 2012 also identified an association between the *ACE* I/D

  

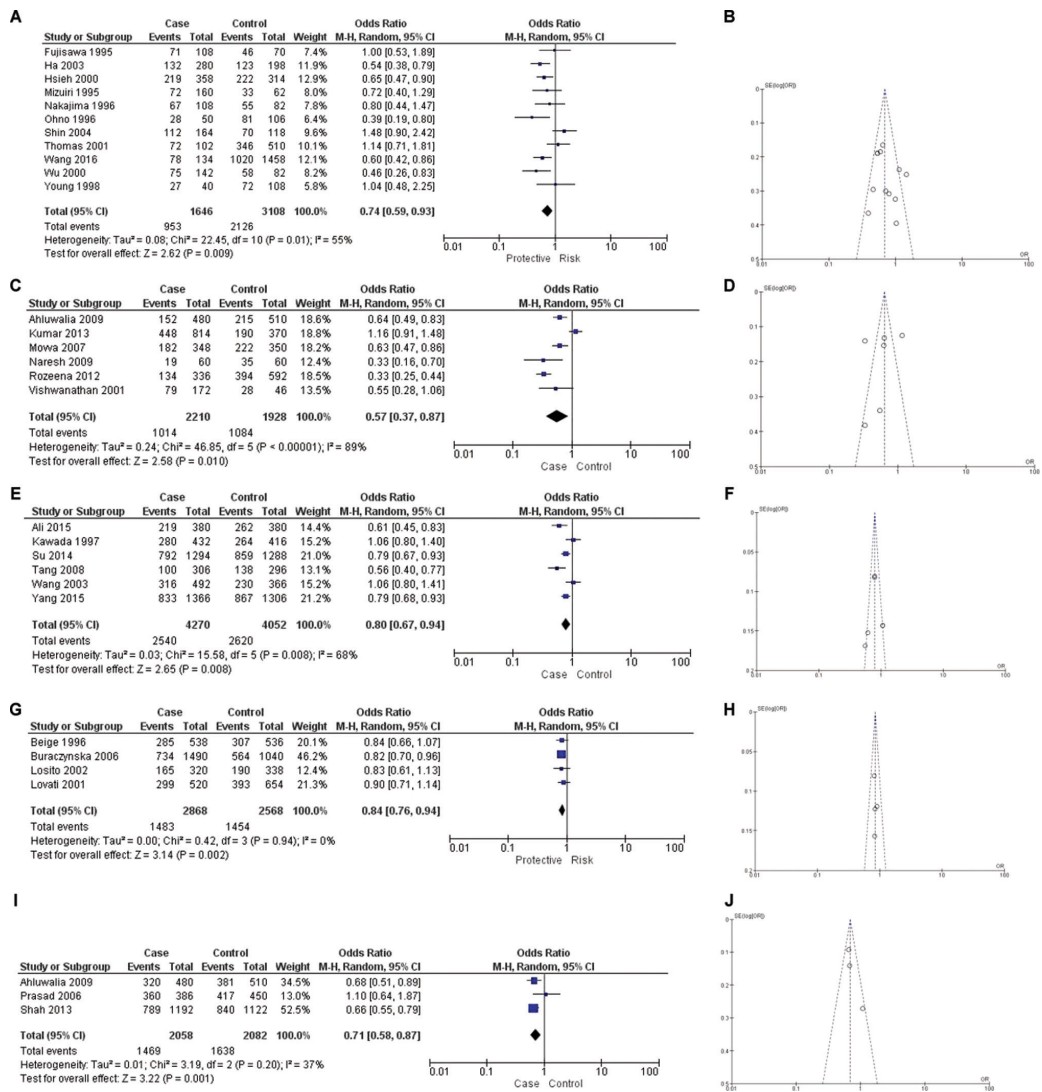

**Figure 2** Forest and funnel plots for statistically significant results. (A) Forest plot—ACE insertion/deletion (I/D) investigation, individuals with type 2 diabetic nephropathy compared with individuals with type 2 diabetes mellitus in an East Asian population (ACE insertion compared with deletion). (B) Funnel plot—ACE I/D investigation, individuals with type 2 diabetes and nephropathy compared with individuals with type 2 diabetes mellitus (without nephropathy) in an East Asian population (ACE insertion compared with deletion). (C) Forest plot—ACE I/D investigation, individuals with type 2 diabetes and nephropathy compared with individuals with type 2 diabetes mellitus (without nephropathy) in a South Asian population (ACE insertion compared with deletion). (D) Funnel plot—ACE I/D investigation, individuals with type 2 diabetic nephropathy compared with individuals with type 2 diabetes mellitus in a South Asian population (ACE insertion compared with deletion). (E) Forest plot—ACE I/D investigation, individuals with end-stage renal disease compared with healthy controls in an East Asian population (ACE insertion compared with deletion). (F) Funnel plot—ACE I/D investigation, individuals with end-stage renal disease compared with healthy controls in an East Asian population (ACE insertion compared with deletion). (G) Forest plot—angiotensinogen (AGT) rs699 investigation, individuals with end-stage renal disease compared with healthy controls in a European population (AGT rs699 T allele compared with C allele). (H) Funnel plot—AGT rs699 investigation, individuals with end-stage renal disease compared with healthy controls in a European population (AGT rs699 T allele compared with C allele). (I) Forest plot—angiotensin II receptor type 1 (AGTR1) rs5186 investigation, individuals with type 2 diabetic nephropathy compared with individuals with type 2 diabetes mellitus in a South Asian population (AGTR1 rs5186 A allele compared with C allele). (J) Funnel plot—AGTR1 rs5186 investigation, individuals with type 2 diabetes and nephropathy compared with individuals with type 2 diabetes mellitus (without nephropathy) in a South Asian population (AGTR1 rs5186 A allele compared with C allele).

polymorphism and the development of DN to ESRD, in that the presence of the deletion was associated with ESRD susceptibility.[36] Despite having differences in the inclusion criteria to our meta-analysis, in that these investigations did not include any article published since 2011[35 36] and included individuals with microalbuminuria

as cases,[35] we returned similar results to the 2005 previous meta-analysis. Including individuals with microalbuminuria in the control population may cause challenges with phenotype definition as microalbuminuria may regress, remain stable or progress to macroalbuminuria[37–39] over time. Individuals with microalbuminuria were therefore

excluded from both case and control definitions to clarify the phenotypes in our review.

In our meta-analysis of *ACE*, which comprised 15 265 individuals with CKD and 18 474 individuals as controls from 98 population groups, we identified three significant associations of the *ACE* I/D polymorphism with CKD. Due to previously reported heterogeneity between different ethnic groups,[35] we included this as a risk factor and performed each analysis per ethnicity (see online supplementary table S2). Despite some of the allele frequency distributions varying across different ethnicities, the direction of the effect was consistent among the different ethnicities: where populations had different minor allele frequencies (MAFs), they often reported similar ORs in the meta-analysis.

First, comparisons between T2DN and T2DM in East Asian and South Asian populations returned significant results highlighting a protective effect of the *ACE* insertion in the development of DN (p=0.009 and p=0.01, respectively). This result was mimicked in the comparison of individuals with ESRD, which was not caused by DN, and healthy control individuals in an East Asian population (p=0.008). The *ACE* I/D polymorphism remains a well-characterised genetic locus associated with the progression of DN.

*AGT* encodes the AGT glycoprotein, which is created in the liver and facilitates the creation of angiotensin I.[11 40] It is located on chromosome 1. Several investigations have been conducted into the *AGT* gene variants and their association with risk of CKD.[40–43] Among these, Zhou and colleagues undertook a meta-analysis investigation into *AGT* rs699 and its association with ESRD.[40] The results of this study are in agreement with ours in relation to the European ethnicity. Our meta-analysis encompassing 5463 individuals with renal disease and 6385 individuals without; the T allele provided a protective effect in ESRD development (p=0.002) within this European population.

In European and Middle Eastern populations, the allele distribution of rs699 was very similar to the comparison of T2DN versus T2DM, yet different from the East Asian population. This may have had an impact on the results, potentially limiting the robustness of association analyses across ethnicities.

The *AGTR1* gene has been shown to be involved in the regulation of blood pressure, fluid and electrolyte balance.[11] It may also have a role in inflammation and vasoconstriction.[44] SNP rs5186, an AC nucleotide substitution at position 1166 in the 3′ untranslated region of chromosome 3, is reportedly able to be recognised by microRNA-155. When the A allele is present at this locus, microRNA-155 is able to undergo complementary base pairing with AGTR1 messenger RNA to suppress translation. However, this is not possible when the alternative C allele is present, resulting in increased AGTR1 protein levels.[45] This interplay may affect blood pressure regulation and warrants further investigation.[45]

Previous studies have identified associations between rs5186 in *AGTR1* and diseases including coronary artery disease,[46] systemic lupus erythematosus[47] and cancer.[48] Several smaller studies had also been undertaken to assess links between this gene and renal disease.[25 49–54] We conducted this study to provide a clearer understanding of the effect of this SNP on CKD.

This meta-analysis of *AGTR1* variants included 3197 individuals with renal disease and 3720 controls investigating rs5186. One significant result was identified: that the presence of an A allele at this locus provided a lower risk of developing T2DN in a South Asian population (p=0.001).

One meta-analysis published in 2014[55] assessed AGTR1 in individuals with CKD, ESRD, IgA nephropathy or vesicoureteral reflux. This meta-analysis identified that rs5186 was not associated with any of these diseases, which corresponds to the results from our investigation.

The three remaining RAAS genes included in this meta-analysis, *ACE2*, *AGTR2* and *REN*, have not been researched as extensively as *ACE*, *AGT* and *AGTR1* for associations with renal disease. Very few articles were identified describing genetic association studies for these genes at the data extraction stage of the analysis, and those that were removed prior to the quantitative analysis stage, mainly due to the inclusion of paediatric individuals or non-human approaches, as outlined in online supplementary figure S1B,E,F. Further research into the *ACE2*, *AGTR2* and *REN* genes and their polymorphisms should be undertaken to elucidate their role in CKD and ESRD.

Some studies in our search could not be included in quantitative analysis as they lacked information relating to genotype counts and had an unclear measure and definition of albuminuria for both cases and controls, which could constitute a limitation. Results from genome-wide association studies would have strengthened the analysis, but unfortunately they usually only report significant SNPs. Absolute frequencies are not usually reported at individual SNP level for such large-scale studies. Publication bias was reported in two of our quantitative analyses, but was not found in the analyses which provided our significant results. Lack of clarity in phenotype definitions, along with unclear descriptions of ethnicities, inherently challenge the use of meta-analyses of different populations as a valid instrument to uncover robust associations. CKD itself has a range of causes including glomerular damage and declining estimated GFR without albuminuria. Other confounding factors such as hypertension and cardiovascular disease, and a lack of prospective follow-up of included individuals, which would ensure phenotypes are robust and stable, may also cause conflicting results.

## CONCLUSION

This meta-analysis of the RAAS pathway genes and their association with renal disease has provided evidence for five significant associations with individually small effect sizes that may cumulatively contribute to dysfunction

of the RAAS pathway leading to kidney disease. The insertion in *ACE* I/D polymorphism was a protection factor for the development of DN in individuals with type 2 diabetes mellitus from both East and South Asian origin, and for ESRD in an East Asian population. In Europeans, the T allele of the *AGT* rs699 conferred a lower risk of ESRD development in healthy population. The A allele in *AGTR1* rs5186 acted as a protection factor for renal disease development in South Asian population.

Further study into the specific ethnicities and investigations into a broader range of RAAS-linked genes, or a deeper analysis of them including investigations of more variants, may pinpoint the molecular basis underlying the role of pathway in kidney disease. Network analysis and functional studies enlightening the mechanisms involved ultimately will be required to complete the picture of RAAS variation in renal traits.

**Contributors** LJS formulated the research plan, conducted the analysis, interpreted the data, drafted and revised the manuscript. MCG and RCC conducted the analysis, interpreted the data and revised the manuscript. APM acquired the funding, interpreted the data, managed the project and revised the manuscript. AM acquired the funding, formulated the research plan, interpreted the data, managed the project and revised the manuscript. All authors read and approved the final manuscript.

**Funding** This work has been partly funded by the Medical Research Council (Award Reference MC_PC_15025) and the Public Health Agency R&D Division (Award Reference STL/4760/13). LJS is the recipient of a Northern Ireland Kidney Research Fund Fellowship. MCG is funded by a Science Foundation Ireland-Department for the Economy (SFI-DfE) Investigator Program Partnership Award (15/IA/3152). RCC is the recipient of a Department for the Economy (DfE) PhD Studentship.

**Competing interests** None declared.

**Patient consent for publication** Not required.

**Provenance and peer review** Not commissioned; externally peer reviewed.

**Data sharing statement** The data sets generated and/or analysed during the current study are available from the corresponding author on reasonable request.

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
