## [Reviewer comments · BMJ Open]

ARTICLE DETAILS

TITLE (PROVISIONAL)	Genetic associations between genes in the renin-angiotensin-aldosterone system and renal disease: a systematic review and meta-analysis
AUTHORS	Smyth, Laura; Cañadas-Garre, Marisa; Cappa, Ruaidhri; Maxwell, Alexander; McKnight, Amy

VERSION 1 - REVIEW

REVIEWER	Dr. L.V.K.S. bhaskar Sickle Cell Institute Chhattisgarh, Raipur, India
REVIEW RETURNED	22-Oct-2018

GENERAL COMMENTS	The manuscript entitled “Genetic associations between genes in the renin- angiotensin-aldosterone system and renal disease: a systematic review and meta-analysis ”, has analysed RAAS genes; ACE, ACE2, AGT, AGTR1, AGTR2 and REN, for their association with CKD. The authors conducted a meta-analysis. My comments were given below. Major comments: The search strategy of the meta-analysis should be more clarified for others to repeat the results. Have the authors performed a manual review of the references in the eligible articles? As increased RAAS activation is linked to progression of CKD of different aetiologies, the authors tried to associate the RAAS gene polymorphisms with the CKD in various diseases. The authors are not cautious and accurate with respect to phenotype description used in the study. Hence I suggest the authors to concentrate their meta-analysis on CKD induced by one disease condition. This will reduce the heterogeneity among the studies. Funnel plot format should be included to facilitate interpretation. Stratified analysis using ethnicity and disease condition should be included. The authors should discuss more about the variations on allele frequency across different studies and assess the effect of the variations on the results.
--

REVIEWER	H Ainsworth, PhD Candidate Wake Forest School of Medicine, USA
REVIEW RETURNED	24-Nov-2018

GENERAL COMMENTS	In the Article, “Genetic associations between genes in the renin-angiotensin-aldosterone system and renal disease: a systemic review and meta-analysis”, the authors undertake a review of chronic kidney disease literature to perform a meta-analysis on summary statistics (allele frequencies, versus raw data) of genetic variants in the RAAS pathway. Strengths:*****  -Very clear hypothesis and objective to specifically evaluate genes related to the RAAS system with evidence from past studies to support their objectives. While individual studies have reported associations with RAAS and CKD, this manuscript aims to compare and analyze variants across these studies. -A comprehensive literature analysis was conducted. The authors took great care to evaluate the biological inclusion criteria of each study and adhered to strict phenotypic definitions (i.e. excluding pediatric cases and individuals with microalbuminuria). -Given ancestral heterogeneity in genetics (linkage disequilibrium) and chronic kidney, the authors analyzed results, separately by ancestry. -For analyses, the authors assumed heterogeneity and appropriately used a random effects model. -The article is well-written and content is clearly organized. Weaknesses:*****  -Given that it's nearly 2019, it's unfortunate that the data curation ends in 2016. Please include recent literature to ensure additional studies meeting inclusion criteria are captured. -In Supplementary Table 1, the search terms indicate that papers were curated by requiring the gene name to be in the search results (i.e. requiring ACE and kidney and SNP). This method would be successful in identifying relevant candidate gene studies. In a candidate gene study, even if the gene of interest is not significant, its name (i.e. ACE) would likely appear in the methods. However, in genome-wide association studies (GWAS), it is likely that *only* the most significantly associated genes would be mentioned in the main text. Additional results would likely be limited to supplemental materials, or not reported at all. Thus, curating literature by requiring the search results to have the name of a particular gene would likely bias to only significant GWAS results. How is this bias from GWAS studies being addressed? The authors should at least include this in the limitations. -While the 0.05 p-value threshold is commonly used among systemic review and meta-analyses in epidemiological studies, it should be in conjunction with correction for multiple comparisons, which was not described in the current manuscript. The authors should apply a method or address this in the discussion, describing why no multiple comparison method is justified. Additional Suggestions:*****  -It would be helpful to better highlight the reference allele and allele frequencies within primary tables. For example, within Table 2, for rs699, which allele corresponds to the 0.84 odds ratio?
--

	Although it's possible for the reader to refer to the supplementary table and tally the counts for the T and C alleles, providing this information (reference alleles and frequencies) within the main results is important to understand the direction of association ("protective" or risk) for a given allele. -The methods state that departure from Hardy Weinberg Equilibrium was calculated for each SNP (line 127) and that is was used in assessing study quality (line 132-133). It is unclear if HWE was calculated by case/control status, or if it was calculated for combined cases and controls. It is possible that a disease variant could be out of HWE in only cases (and in HWE for controls). -For the analyzed variants, do the control allele frequencies appear to match what is expected in the matched population (i.e. 1000 genomes data). This would reduce a concern that there might be bias in the published associations from the null.
--	--

VERSION 1 – AUTHOR RESPONSE

Response to Reviewer 1 Comments

We would like to take this opportunity to thank the Reviewer for their comments and suggestions for this article. Changes have been made to strengthen this manuscript and address all comments as outlined below:

Please leave your comments for the authors below

The manuscript entitled "Genetic associations between genes in the renin- angiotensin-aldosterone system and renal disease: a systematic review and meta-analysis", has analysed RAAS genes; ACE, ACE2, AGT, AGTR1, AGTR2 and REN, for their association with CKD.

The authors conducted a meta-analysis.

My comments were given below.

Major comments:

1. The search strategy of the meta-analysis should be more clarified for others to repeat the results. Have the authors performed a manual review of the references in the eligible articles?

We have further clarified the search strategy of this meta-analysis and included additional information within the Search Strategy sub-section of Methods. We have performed a manual review of the references in each of the eligible articles and have stated this in the Search Strategy sub-section of Methods – page 3, lines 100-106 (tracked changes document).

2. As increased RAAS activation is linked to progression of CKD of different aetiologies, the authors tried to associate the RAAS gene polymorphisms with the CKD in various diseases.

The authors are not cautious and accurate with respect to phenotype description used in the study. Hence I suggest the authors to concentrate their meta-analysis on CKD induced by one disease condition. This will reduce the heterogeneity among the studies.

To reduce heterogeneity, studies which focused on CKD progression were excluded from the analysis and we have now stated this in the Methods – page 5, lines 142-145 (tracked changes document).

The analysis was stratified by disease type and control individuals – we excluded any published article which did not provide a clear definition of the clinical phenotype of the underlying kidney disease. The comparisons shown in Table 1 (pages 3 and 4) show that we did not assume homogeneity for any phenotype – if the definition was not clear, a separate group was created. For example: Individuals presented with ESRD derived from diabetes, but on some occasions the type of diabetes was not explicitly stated in cases or controls, therefore three groups were created – T1-ESRD/T1DM, T2-ESRD/T2DM, DM-ESRD/T1DM, and were analysed separately.

3. Funnel plot format should be included to facilitate interpretation.

The original funnel plots have been replaced by those which include the 95% confidence indication lines – these have been included in the Supplementary Figures PDF.

4. Stratified analysis using ethnicity and disease condition should be included. The authors should discuss more about the variations on allele frequency across different studies and assess the effect of the variations on the results.

Ethnicity and disease stratification were taken into consideration and have been included in the manuscript. We have further discussed the effect of the different weights and direction of effects from each meta-analysis in the Results section of the text. Regarding the allele frequency variation, they were assessed in controls within each meta-analysis and were not statistically different in any of the stratified analyses (data not shown).

Despite some of the allele frequency distributions varying across different ethnicities, the direction of the effect was consistent among the different ethnicities: where populations had different MAFs, they often reported similar odds ratios in the meta-analysis. We carefully considered this complication using sub-group analysis for individual ethnicities.

We have included this in the Discussion, page 10, lines 286-288.

Only in the example of AGT: T2DN vs T2DM comparison, may different MAFs across ethnicities have had an impact on the results.

EAS OR 0.69

EUR OR 1.03

ME OR 1.19

In European (EUR) and Middle Eastern (ME) populations, the allele distributions were very similar, yet different from the East Asian (EAS) population.

We have discussed this important aspect in the Discussion, page 11, lines 302-304 (tracked changes document).

Response to Reviewer 2 Comments

We would like to take this opportunity to thank the Reviewer for their comments and helpful suggestions for this article. Changes have been made to strengthen this manuscript and address all comments as outlined below:

Please leave your comments for the authors below

In the Article, “Genetic associations between genes in the renin-angiotensin-aldosterone system and renal disease: a systemic review and meta-analysis”, the authors undertake a review of chronic

kidney disease literature to perform a meta-analysis on summary statistics (allele frequencies, versus raw data) of genetic variants in the RAAS pathway.

Strengths:

1. Very clear hypothesis and objective to specifically evaluate genes related to the RAAS system with evidence from past studies to support their objectives. While individual studies have reported associations with RAAS and CKD, this manuscript aims to compare and analyze variants across these studies.
2. A comprehensive literature analysis was conducted. The authors took great care to evaluate the biological inclusion criteria of each study and adhered to strict phenotypic definitions (i.e. excluding pediatric cases and individuals with microalbuminuria).
3. Given ancestral heterogeneity in genetics (linkage disequilibrium) and chronic kidney, the authors analyzed results, separately by ancestry.
4. For analyses, the authors assumed heterogeneity and appropriately used a random effects model.
5. The article is well-written and content is clearly organized.

Weaknesses:

1. Given that it's nearly 2019, it's unfortunate that the data curation ends in 2016. Please include recent literature to ensure additional studies meeting inclusion criteria are captured.

We have updated the search to include all articles published prior to 01/01/2019 and have updated the Abstract, page 1, line 31 and Methods section, page 3, line 98 (tracked changes document) to reflect this.

2. In Supplementary Table 1, the search terms indicate that papers were curated by requiring the gene name to be in the search results (i.e. requiring ACE and kidney and SNP). This method would be successful in identifying relevant candidate gene studies. In a candidate gene study, even if the gene of interest is not significant, its name (i.e. ACE) would likely appear in the methods. However, in genome-wide association studies (GWAS), it is likely that *only* the most significantly associated genes would be mentioned in the main text. Additional results would likely be limited to supplemental materials, or not reported at all. Thus, curating literature by requiring the search results to have the name of a particular gene would likely bias to only significant GWAS results. How is this bias from GWAS studies being addressed? The authors should at least include this in the limitations.

We have included a statement in the Discussion to highlight the limitation of potential GWAS associations – page 11, paragraph 6, lines 332-334 (tracked changes document).

3. While the 0.05 p-value threshold is commonly used among systemic review and meta-analyses in epidemiological studies, it should be in conjunction with correction for multiple comparisons, which was not described the current manuscript. The authors should apply a method or address this in the discussion, describing why no multiple comparison method is justified.

Multiple comparisons have now been applied – taking into consideration the SNPs analysed for each ethnicity and phenotype.

	T2DN, SA	T2DN, EA	ESRD, EUR	ESRD, EA
ACE	0.01	0.009	0.86	0.008
AGT	*Excluded	0.12	0.002	0.89
AGTR1	0.001	0.71	*Excluded	*Excluded
Corrected p value	0.025	0.017	0.025	0.025

*Excluded as only two populations were available and therefore was not included in the meta-analysis.

No variations in the significant results after considering the Bonferroni-corrected p-value. We would like to restate that the significant SNPs identified in this meta-analysis were replicated across multiple studies, with the same direction of effect.

Additional Suggestions

4. It would be helpful to better highlight the reference allele and allele frequencies within primary tables. For example, within Table 2, for rs699, which allele corresponds to the 0.84 odds ratio? Although it's possible for the reader to refer to the supplementary table and tally the counts for the T and C alleles, providing this information (reference alleles and frequencies) within the main results is important to understand the direction of association ("protective" or risk) for a given allele.

We have updated Table 2, page 6 of the tracked changes document, and provided additional information within this primary table regarding the allele frequencies. Additional allele frequency data is available in the Supplementary Data File S1.

5. The methods state that departure from Hardy Weinberg Equilibrium was calculated for each SNP (line 127) and that is was used in assessing study quality (line 132-133). It is unclear if HWE was calculated by case/control status, or if it was calculated for combined cases and controls. It is possible that a disease variant could be out of HWE in only cases (and in HWE for controls).

The Hardy Weinberg Equilibrium calculations were completed for cases and controls individually. We have updated the Methods text to state this, page 5, lines 134 and 135.

6. For the analyzed variants, do the control allele frequencies appear to match what is expected in the matched population (i.e. 1000 genomes data). This would reduce a concern that there might be bias in the published associations from the null.

We have checked the MAFs available from dbSNP and have compared these with any meta-analysis conducted where the control population consisted of healthy individuals for both AGT rs699 and AGTR1 rs5186. None of the SNP rs numbers (rs13447447, rs1799752, rs4340, rs4646994) previously reported for the ACE insertion/deletion are included within dbSNP – the SNPs had either been withdrawn from dbSNP, or no frequencies were available.

We have included information regarding the MAFs for rs699 and rs5186 in the results section of the manuscript - page 9, lines 251-254 and Supplementary Data File S1.